# Changes in Endothelial Nitric Oxide Production in Systemic Vessels during Early Ontogenesis—A Key Mechanism for the Perinatal Adaptation of the Circulatory System

**DOI:** 10.3390/ijms20061421

**Published:** 2019-03-21

**Authors:** Dina K. Gaynullina, Rudolf Schubert, Olga S. Tarasova

**Affiliations:** 1Faculty of Biology, M.V. Lomonosov Moscow State University, Moscow 119234, Russia; ost.msu@gmail.com; 2Department of Physiology, Russian National Research Medical University, Moscow 117997, Russia; 3Centre for Biomedicine and Medical Technology Mannheim (CBTM) and European Center of Angioscience (ECAS), Research Division Cardiovascular Physiology, Medical Faculty Mannheim, Heidelberg University, 68167 Mannheim, Germany; Rudolf.Schubert@medma.uni-heidelberg.de; 4Department of Physiology, Medical Faculty, Augsburg University, 86159 Augsburg, Germany; 5State Research Center of the Russian Federation—Institute for Biomedical Problems, Russian Academy of Sciences, Moscow 123007, Russia

**Keywords:** Nitric oxide, eNOS, ontogenesis, postnatal development, thyroid hormones, hypothyroidism, circulatory system

## Abstract

Nitric oxide (NO) produced in the wall of blood vessels is necessary for the regulation of vascular tone to ensure an adequate blood supply of organs and tissues. In this review, we present evidence that the functioning of endothelial NO-synthase (eNOS) changes considerably during postnatal maturation. Alterations in NO-ergic vasoregulation in early ontogeny vary between vascular beds and correlate with the functional reorganization of a particular organ. Importantly, the anticontractile effect of NO can be an important mechanism responsible for the protectively low blood pressure in the immature circulatory system. The activity of eNOS is regulated by a number of hormones, including thyroid hormones which are key regulators of the perinatal developmental processes. Maternal thyroid hormone deficiency suppresses the anticontractile effect of NO at perinatal age. Such alterations disturb perinatal cardiovascular homeostasis and lead to delayed occurring cardiovascular pathologies in adulthood. The newly discovered role of thyroid hormones may have broad implications in cardiovascular medicine, considering the extremely high prevalence of maternal hypothyroidism in human society.

## 1. Endothelium-Derived Nitric Oxide (NO) Controls Vascular Tone

The endothelium is a cellular monolayer that lines the inside of all blood and lymphatic vessels and is characterized by a high secretory activity. Endothelial cells produce a variety of vasoactive substances, the action of which is normally aimed at reducing the hydrodynamic resistance of blood vessels. Nitric oxide (NO) is one of the major relaxing factors produced by the endothelium [1]. NO is synthesized by NO-synthase, represented by endothelial, neuronal and inducible isoforms. The endothelial isoform of the enzyme (eNOS) is the most common in the healthy vascular system [1].

The regulation of eNOS activity in endothelial cells is carried out in several ways. In the long term, its activity depends on the content of the substrate L-arginine and cofactors (primarily BH_4_) as well as endogenous inhibitors (caveolin, ADMA, etc.) [1,2]. A relatively rapid change in eNOS activity occurs in response to an alteration in the cytoplasmic Ca^2+^ concentration in endothelial cells or during eNOS phosphorylation at different regulatory sites as a result of the effect of vasoactive substances or the action of shear stress. Phosphorylation of eNOS can be carried out by various protein kinases, modulating eNOS activity directly or by changing its interaction with other regulatory proteins [2]. Among the phosphorylation sites, the roles of the activation site serine-1177 (a target of Akt, PKA, AMPK, CaMKKII beta etc.) and the inhibitory site threonine-495 (a target of Rho-kinase and protein kinase C) are the most established [3].

## 2. The Vascular System in Early Postnatal Ontogenesis

Early postnatal ontogenesis is characterized by intensive growth and development of tissues and organs, including the cardiovascular system. In the period after birth, the circulatory system undergoes considerable remodeling at the structural (such as an increase in the length and number of vessels, as well as their wall thickness) and the functional (for example, a decrease in the calcium sensitivity of the contractile apparatus) levels.

At the systemic level, a distinctive feature of the healthy newborn organism is a low blood-pressure level, which increases during the first months of life in humans and in the first weeks after birth in rodents [4,5]. In addition, during early postnatal ontogenesis, the distribution of cardiac output between various organs is very different to the adult organism: blood supply to the brain, skeletal muscles and kidneys is lower, and that to the intestine and the skin is higher than in adults [6]. The redistribution of blood flow between different organs in postnatal ontogenesis is due to a change in a variety of mechanisms that regulate vascular tone, including the postnatal maturation of sympathetic nervous control [7,8], as well as changes in the functional activity of smooth muscle cells [5,8,9,10]. In addition, the observation of diverse blood flow alterations during the postnatal period in different organs of the systemic circulation suggests the existence of organ-specific postnatal changes in the vasodilatory effects of endothelium-derived NO.

### 2.1. Organ-Specific Developmental Alterations of NO-Mediated Vascular Control

#### 2.1.1. Cerebral Circulation

An adequate cerebral blood flow is the key task of the cardiovascular system at all stages of ontogenesis. Blood supply to the brain was shown to increase considerably from the perinatal period to adulthood [6,11]. The vascular endothelium plays an important role in the maturational rise in blood flow. 

Endothelium-dependent relaxation is getting stronger during postnatal ontogenesis in cerebral vessels of pigs [12], sheep [13,14] and mice [15,16]. Moreover, cerebral arteries of newborn animals even demonstrate an endothelium-dependent contraction associated with the synthesis of endothelin-1, which disappears with maturation [17]. 

Importantly, with maturation of the organism, there can be a switch of the mechanisms providing endothelium-dependent relaxation. For example, a greater contribution of prostanoids to endothelium-dependent relaxations of cerebral vessels was observed in newborn pigs, whereas the relaxation becomes more dependent on NO synthesis in adulthood [18,19,20]. The increase of the NO component in the endothelium-dependent relaxation with age was also demonstrated in cerebral arteries of sheep, which, however, do not depend on prostanoid synthesis in any age group [14]. 

The maturational increase in the contribution of NO to endothelium-dependent relaxation of cerebral arteries is not associated with an increase in the sensitivity of arterial smooth muscle to NO during ontogenesis, as the responses of these arteries to NO donors do not change with age [15,16,19]. This emphasizes the endothelial origin of the enhanced contribution of NO to vascular tone regulation during postnatal development. Indeed, the increase of the NO component in the endothelium-dependent relaxation of these arteries during early ontogenesis is accompanied either by an increase of eNOS protein content in the cerebral arteries, or by an increase of eNOS activity [21]. The increase in eNOS activity is due to its elevated phosphorylation at serine-1177 [16]. Along with that, a decrease in the inhibitory effect of endothelial Rho-kinase was observed in cerebral arteries from mice during development [15].

Taken together, we conclude that an increase in endothelium-dependent relaxation of cerebral arteries and arterioles occurs during postnatal ontogenesis. This increase can be accompanied by a switch in the mechanisms governing endothelium-dependent relaxation from more prostanoid-dependent to more NO-dependent ones. 

#### 2.1.2. Coronary Circulation

The heart is highly metabolically active during the whole lifespan. Therefore, appropriate delivery of blood through the coronary vasculature is vitally important. In the neonatal pig coronary circulation, NO was shown to exert a powerful anticontractile effect on basal tone and to mediate a substantial part of agonist-stimulated dilatory responses [22]. In the neonatal rat heart endogenous NO is protective against ischemia/reperfusion injury [23]. Of note, in coronary arteries from adult rats a strong augmentation of basal tone and contractile responses was demonstrated after treatment with a NOS inhibitor [24]. To our knowledge, data reporting a direct comparison of vasomotor NO effects between newborn and adult individuals are not available. We propose that a strong NO-dependent control of coronary artery tone is essential for an adequate blood supply to the heart throughout life. This is supported by data showing an unchanged coronary blood flow in rats between the 9th and 64th days of postnatal life [6] but has to be confirmed by future studies.

#### 2.1.3. Skeletal Muscle Circulation

During early postnatal ontogenesis, growth and development of skeletal muscles occurs, coupled to an increase of the locomotor activity of the individual. This, in turn, entails the need for a more intense blood supply to skeletal muscles. As in other vascular beds, this change in blood flow is associated, among others, with changes in the endothelium-dependent regulation of vascular tone of skeletal muscle arteries [25,26]. Studies addressing endothelium-dependent mechanisms of tone regulation in the skeletal muscle vasculature have been performed predominantly on rats and can be classified into two groups. The first group of studies involves skeletal muscle arterioles, while the second group considers larger skeletal muscle feed arteries.

In rat skeletal muscle arterioles endothelium-dependent relaxations have been shown not to change during juvenile growth [25] or even to decline with age [26], depending on the particular muscle group. Importantly, the contribution of NO to these endothelium-dependent relaxations increases with age [26]. Similarly, the anticontractile effect of NO has been shown to increase during juvenile growth in skeletal muscle arterioles [25]. Interestingly, the responses of skeletal muscle arterioles upon activation of endothelial Ca^2+^-dependent pathways inducing vasorelaxation do not differ between 4- to 5- and 7- to 8-week-old rats. However, vasodilator stimuli associated with eNOS phosphorylation induced NO production in the older but not in the younger age group [27]. In addition, in the older age group, the level of eNOS phosphorylation at serine-1177 is higher, and the level of phosphorylation at threonine-495 is lower compared to the younger age group. This indicates a greater degree of eNOS activation in older animals, despite a similar eNOS protein content [28]. In contrast to arterioles, a decrease of the anticontractile effect of NO with maturation has been shown in larger skeletal muscle arteries from rats. Namely, in the gastrocnemius muscle feed artery the effect of an eNOS inhibitor gradually disappeared by the age of 10–12 weeks [29]. Qualitatively similar data were obtained for the larger popliteal artery [30]. The maturational decrease in the effect of NO was associated with an increase in the content of arginase-2, an enzyme competing for the substrate with eNOS [29]. 

Thus, the role of NO in the regulation of skeletal muscle vascular tone during early postnatal ontogenesis increases in arterioles but decreases in larger feed arteries.

#### 2.1.4. Renal Circulation

The kidney of neonates is characterized by a high vascular resistance, as well as by a low blood flow and glomerular filtration rate compared with the kidney of an adult organism [6,31]. Consistent with this, the anticontractile influence of NO is absent in large interlobar arteries of 2-week-old rats, while it reduces arterial tone in adult rats [29]. 

Along with that, in small preglomerular microvessels the vasodilatory influence of NO on the vascular tone of newborn piglets is higher compared to adult animals, but this effect is due to the activity of the neuronal isoform of NO-synthase [32], leading to a reduction of microvascular resistance due to tonic anticontractile influence. Of note, the expression level of the neuronal isoform of NO-synthase in these microvessels gradually decreases from a maximum at birth to adult age [33,34]. The authors of these studies suggest that a high level of NO production in newborn piglets is protective and the vasodilatory effect of NO counteracts the high activity of the renin-angiotensin system during early postnatal development, thereby protecting the kidney from damage.

Thus, the contribution of NO to the regulation of the renal circulation during early postnatal ontogenesis increases in larger arteries but decreases in preglomerular microvessels. However, the latter effect is not dependent on the endothelial isoform of NO-synthase.

#### 2.1.5. Intestinal Circulation

In mammals during the prenatal period of life nutrition is enabled by the placenta, therefore the gastrointestinal tract of the fetus is inactive. However, the situation changes after birth, when a switch to nutrition through the gastrointestinal tract takes place. A rapid increase in body weight after birth becomes possible due to active food consumption and digestion. Therefore, the functioning of the gastrointestinal tract in early postnatal ontogenesis is accompanied by an intense blood supply. In pigs, intestinal blood flow demonstrates biphasic changes during the first month of life with a high level of blood flow on the first day of life, which increases even more by day 3 of postnatal ontogenesis and then gradually declines [35]. Such alterations in blood supply to the gastrointestinal tract are provided, in particular, by changes in the endothelium-dependent regulation of arterial tone.

The endothelium-dependent relaxation in the swine mesenteric vasculature has been shown to decrease during the first month of postnatal development [36]. A similar decline was demonstrated for the NO-mediated anticontractile effect on mesenteric resistance arteries [37]. The level of eNOS protein expression and the shear stress-induced NO synthesis are higher in the mesenteric vasculature of neonatal pigs compared to 1-month-old animals [38]. 

During the subsequent postnatal ontogenesis, the anticontractile influence of NO gradually decreases [29]. Interestingly, this decrease in the role of NO in the regulation of arterial contractile responses in rats with age is accompanied by an increase in arginase-2 levels but not associated with changes in the expression of eNOS mRNA [29]. In addition, the NO-dependent mechanisms of mesenteric artery smooth muscle relaxation change with age: the expression of the regulatory subunit of myosin light chain phosphatase switches from a more NO-sensitive to a less sensitive isoform [39], although the sensitivity of arterial smooth muscle to exogenous NO does not decrease with age [29].

Thus, the presented data suggest that, after an early increase, a decrease in the contribution of NO to the regulation of vascular tone in the intestine takes place during early postnatal development.

#### 2.1.6. Cutaneous Circulation

Skin blood flow undergoes considerable changes during postnatal ontogenesis: it accounts for up to 20% of cardiac output in 1- to 2-week-old rats but decreases considerably at adult age [6]. Accordingly, the regulation of cutaneous blood flow, including the NO-dependent component, is considerably altered during development. The endothelium of a skin feed artery (namely, the a. saphena) of 1- to 2-week-old rats has been shown to produce NO, which has a powerful anticontractile influence that strongly decreases during maturation [29,30]. This process is associated with a decrease in the eNOS expression level during postnatal ontogenesis, while smooth muscle sensitivity to NO remains unchanged [30]. Thus, the contribution of NO to the regulation of cutaneous circulation decreases during postnatal maturation.

#### 2.1.7. The Impact of NO for Systemic Vascular Control Decreases with Age

Summarizing the above data on changes in the endothelium-dependent regulation of vascular tone during early postnatal ontogenesis, we conclude that such changes are organ-specific and in many cases associated with changes in the contribution of NO to arterial tone regulation (Table 1). The decrease in the contribution of NO as observed in several organs is unlikely to be related to the thickening of the vascular smooth muscle layer during maturation, because vessels in other organs demonstrate opposite changes in the functional role of NO. Of note, the NO-dependent control of vascular tone interacts with other control mechanisms. For example, an intensification of tissue metabolism in such organs as the heart, skeletal muscle and intestine can be beneficial for eNOS activity in, for example, both smaller intramuscular arteries (due to interstitial acidification) [40] and larger feed arteries (due to an increase of shear stress acting on the endothelium) [41]. Of note, changes in tissue metabolism during early postnatal ontogenesis contribute to the adaptation of the vascular system during development. 

At the systemic level during the early postnatal period in rodents as well as in humans the blood pressure level is considerably lower than in adults [4,5,29,42,43]. This low blood pressure level seems to be, at least partially, associated with a higher contribution of NO to arterial tone regulation in a variety of vascular beds at younger age [29]. In accordance with this, a greater rise of arterial blood pressure was observed during NO-synthase inhibition in young rats than in adults [29]. In addition, this correlates with the content of NO metabolites in the blood which is considerably higher at an early age in different mammalian species [29,30,44]. Importantly, a high level of NO metabolites is observed also in the blood of newborn humans, which gradually decreases with maturation, emphasizing the relevance of the described phenomenon for the human organism [45]. 

## 3. Thyroid Hormones and NO

### 3.1. Basics of Thyroid Effects on the Vascular Endothelium

The functional activity of the endothelium, including NO production, is controlled by a variety of hormonal systems, including thyroid hormones [46]. Of note, thyroid hormones are essential for the development of the cardiovascular system [47,48]. Importantly, in many mammalian species the blood concentration of thyroid hormones was shown to be higher during the early stages of development compared to adulthood. For example, a rise in blood triiodothyronine (T_3_) concentration was shown between gestational week 30 and birth in humans and between birth and the 3rd postnatal week in rats [49,50]. Thyroid hormones can have rapid and long-term effects on eNOS expression and/or activity in endothelial cells (Figure 1). 

The functional relationship between thyroid hormones and eNOS on a short time scale was demonstrated at the systemic level: the acute administration of T_3_ leads to a considerable decrease in blood pressure in control mice, but not in eNOS knockout mice [51]. At the level of isolated arteries, the rapid direct vasodilatory effects of thyroid hormones were shown to be mediated by NO [52,53]. The rapid non-genomic action of thyroid hormones in endothelial cells can occur upon binding to the TRα1 receptor, which activates the PI3/Akt-kinase signaling pathway in endothelial cells, causing phosphorylation and activation of eNOS [51]. In addition, another non-genomic pathway is the T_4_ interaction with membrane integrin receptors for thyroid hormones, which causes the activation of PI3-kinase [54], followed by eNOS activation. 

The long-term effects of thyroid hormones on eNOS expression/activity in the endothelium were investigated in experimental models of hyperthyroidism and hypothyroidism. Chronic hyperthyroidism leads to an increase in NO synthesis due to an increase in eNOS protein content [55] or an increase in both eNOS content and activity [56,57] in arterial tissue. Chronic hypothyroidism, in contrast, leads to a decrease in eNOS content [57] and a drop in the content of NO metabolites in the blood [58]. In addition, it has been shown that in people suffering from hypothyroidism, endothelium-dependent relaxations of the forearm vasculature to acetylcholine are decreased due to a reduced production of NO [59]. Similar observations were made for the coronary vasculature of women suffering from hypothyroidism [60]. 

Summarizing, we conclude that thyroid hormones have a stimulating effect on the production of NO by the vascular endothelium. Higher thyroid hormone levels in the early postnatal period compared with the adult organism correlate with the increased NO contribution to the regulation of arterial tone at a younger age. Further, we will discuss the alterations of the functional activity of the endothelium resulting from a disturbed hormonal regulation in the period of early ontogeny. Unfortunately, the problem of hormonal imbalance in early ontogeny is urgent in the modern world: maternal hypothyroidism and, subsequently, the hypothyroidism of fetuses, which depend on maternal thyroid hormones for a considerable part of intrauterine development, accompanies up to 15% of all pregnancies [61].

### 3.2. Maternal Hypothyroidism Weakens NO-Mediated Vascular Tone Control in the Early Postnatal Period

In the early postnatal period, maternal hypothyroidism affects the NO-mediated control of offspring vascular tone in an organ-specific manner. A study of a rat model of antenatal/early postnatal hypothyroidism shows that the anticontractile effect of NO is considerably reduced in mesenteric arteries of hypothyroid offspring at the age of 2 weeks, while it is not altered in their subcutaneous artery (a. saphena) [62]. The diverse sensitivity of the two studied vascular beds to maternal hypothyroidism can be explained by their intrinsic differences in the contribution of NO to the regulation of vascular smooth muscle tone. The NO-mediated anticontractile effect was shown to be considerably more pronounced in the mesenteric compared to the cutaneous vasculature [29,62]. Therefore, the differences in the influence of maternal hypothyroidism on the NO-dependent control of vascular tone in the mesenteric and cutaneous circulation emphasize the peculiarities of different vascular beds and the need for further investigation. 

Evidence for the influence of maternal hypothyroidism on NO-mediated vascular control in early ontogenesis was also obtained at the systemic level. The amount of NO metabolites in the blood of 2-week-old pups from hypothyroid dams is considerably reduced, suggesting that not only the mesenteric, but also other vascular regions can be affected by maternal hypothyroidism [62]. Importantly, a decline in systemic NO production was also reported in human newborns with congenital hypothyroidism [63]. 

### 3.3. Maternal Hypothyroidism Weakens NO-Mediated Vascular Tone Control in Adult Offspring

The accumulated data in the literature indicate that dysregulation of the normal development and functioning of the organism, including the cardiovascular system, during early ontogenesis can have a negative impact on the health status of an individual in adulthood [53]. According to numerous observations, maternal hypothyroidism can cause marked disturbances in the functioning of the cardiovascular system in the adult offspring, despite the restoration of blood thyroid hormone concentrations [24,64,65,66,67,68]. In humans, the delayed effects of maternal hypothyroidism during the third trimester of pregnancy are manifested in their descendants as an increased blood pressure at young (about 20 years) age [65]. This is consistent with experimental data obtained in rats exposed to hypothyroidism from day 9 of gestation to birth: at the age of 2 months, their offspring also have elevated mean, systolic and diastolic blood pressure levels [66]. Notably, in children with congenital hypothyroidism, dysfunction of endothelium-dependent vasodilation is diagnosed in adulthood (about 19 years old) despite hormone replacement therapy provided from the first month of life [69].

Mechanistic studies revealed a strong association between endothelial dysfunction resulting from maternal hypothyroidism and the deficiency of NO-mediated vascular control. The strong anticontractile effect of NO typical for the coronary vasculature is completely abolished in coronary arteries from adult rat offspring of hypothyroid dams [70]. As a result, coronary arteries of hypothyroid dam offspring demonstrated an increase in the level of basal tone and contractile responses to the activation of thromboxane A2 receptors. Importantly, experimental maternal hypothyroidism in rats during pregnancy causes a decrease in the content of NO metabolites in the heart of their adult offspring [71], while the content of NO metabolites in the blood remains unchanged [24,71]. These data indicate again that the effect of maternal hypothyroidism on the NO-dependent regulation of arterial tone in adulthood has regional specificity and the coronary circulation seems to be particularly sensitive to the hormonal imbalance during the early stages of ontogenesis. Importantly, NO produced by the vascular endothelium plays an important role in the coronary circulation [72,73]. The lack of tonic NO action in coronary arteries of previously hypothyroid offspring may be provocative for vasospasms as well as platelet adhesion, smooth muscle growth, inflammatory processes and atherosclerotic plague formation.

## 4. Conclusions

Taken together, the data described above demonstrate that the vasodilator role of NO is augmented in numerous vascular beds in the early postnatal period compared to adult age. Consequently, the influence of NO may be a mechanism which lowers arterial pressure in young individuals. Thyroid hormones participate in the short-term and long-term control of NO production by the vascular endothelium. Maternal hypothyroidism can lead to considerable changes in the NO-mediated control of vascular tone, which may disturb cardiovascular homeostasis in the early postnatal period and have consequences in adulthood. However, considering the extreme clinical importance of this problem and taking into account the pronounced peculiarities of different vascular beds, further studies on this issue should be conducted.

## Figures and Tables

**Figure 1 ijms-20-01421-f001:**
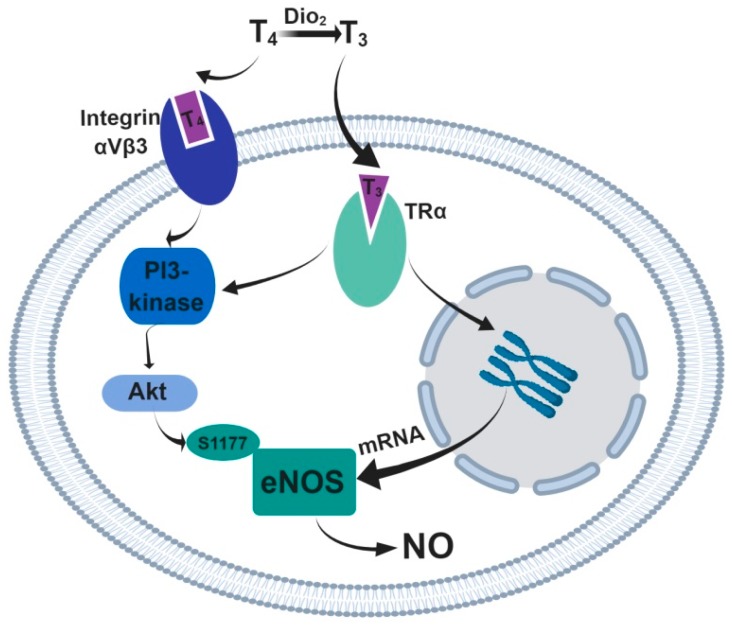
The effects of thyroid hormones on the activity and expression level of eNOS in vascular endothelial cells. Thyroxin (T_4_) is converted to its active form triiodothyronine (T_3_) by deiodinase 2 (Dio_2_). A rapid non-genomic action of thyroid hormones can occur upon binding T_3_ to the TRα and/or T_4_ to integrin receptors, which both can activate the PI3/Akt-kinase signaling pathway, causing phosphorylation at the serine-1177 site and activation of eNOS. The long-term genomic effects of thyroid hormones occur upon binding of T_3_ to TRα receptors, which triggers expression of eNOS mRNA.

**Table 1 ijms-20-01421-t001:** Organ-specific changes in the contribution of endothelial NO-synthase (eNOS)-derived NO to the regulation of vascular tone during early ontogenesis.

Organ	What Happens to the Organ and Its Blood Supply at Birth or in the Early Postnatal Period?	The Contribution of eNOS-derived NO
Brain	Increase of metabolic demand; pronounced growth of cerebral blood flow	Increases with maturation [16,18,19,20,21]
Skeletal muscles	Increase of locomotor activity; increase of blood supply to the muscles	Grows in arterioles with maturation [25,26] but declines in larger arteries [29,30]
Kidney	Branching of the arterial network; increase of blood flow and glomerular filtration rate	Increases with maturation in large arteries [29]
Intestine	Switching to external nutrition immediately after birth; blood flow is high in the first days-weeks of life, then decreases	Most prominent in the early postnatal period, then decreases [29,37,38]
Skin	A significant decrease in the surface/volume ratio of the body at the stage of rapid growth of the organism; decrease in total blood flow in the skin	Most prominent in the early postnatal period, then decreases [29,30]

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
