# Peer review of "Changes in Endothelial Nitric Oxide Production in Systemic Vessels during Early Ontogenesis—A Key Mechanism for the Perinatal Adaptation of the Circulatory System"

_ijms, 2019, doi:10.3390/ijms20061421_

Round 1

Reviewer 1 Report

Summary:  This review evaluates the literature on the ontogenesis of endothelium-derived NO in regulating organ blood flow in different circulations.  Further, the role of thyroid hormone during pregnancy is discussed in the context of its regulation of NO during early neonatal and adult stages of maturation.  It concludes that since thyroid hormone plays an important role in NO production that the alteration in thyroid hormone levels (i.e. during hypo and hyperthyroidism) in pregnant women is an important regulator of organ blood flow and systemic blood pressure in the early vs late offspring.

Strengths – The review is very well written and covers specific topics in a clear and organized manner.  It does a good job of addressing the role of eNOS in organ-specific circulations and the ontogenetic changes that occur in blood vessels as well as the possible underlying mechanisms that contribute to the changes in eNOS activation with age.  The role of thyroid hormone as an important regulator of NO production and, hence, vascular tone in conditions of hypo vs hyperthyroidism is well made and convincing.  Further, the case is well made as to the effect of maternal hypo/hyperthyroidism on programming effects of vascular responses in the offspring.

Concerns to address –

The coronary circulation is an important vascular bed that should be included in a separate paragraph.  It could still be included with regards to the influence of thyroid levels as in the latter sections.  It might be important to address other endothelium-derived factors such as epoxyeicosatrienoic acids as endothelium-derived hyperpolarizing factors, particularly in the coronary circulation.  Lastly, there is a very nice review on the role of thyroid hormone in development of the fetal heart (SS Jonker and S Louey, J Endocrinol., 2016).  While it does not address endothelium-dependent mechanisms it may play into your thoughts regarding thyroid’s developmental role in the coronary circulation.

While the focus of this article is on NO’s role in mediating vascular reactivity, when discussing vascular tone and blood flow and the ontogenetic changes that occur, the role of local tissue metabolism is not mentioned.  It should be included that in certain organs such as skeletal muscle and intestine changes in vascular tone are highly influenced by its metabolic rate and motility, respectively.  Thus, while changes in eNOS activity may contribute to baseline levels of tone, there may be other factors that contribute to ontogenetic changes in blood flow such as tissue metabolism rather than to NO release only. 

Another mechanism of reduced NO influence with age could be related to a maturational increase in the number of vascular smooth muscle layers from early to late adult stages.  Thus, NO may have a lesser influence on artery walls with thicker versus thinner thickness due to the number of layers and a lesser influence on vascular tone.  This was marginally addressed on Line 55 but could be addressed more directly.

Information on thyroid levels would be helpful.  This information, if available, would help put the role of thyroid hormones in a physiological vs pathological context. 

Renal circulation – Line 130 –clarify “contribution of NO to vascular tone” – do you mean in reducing vascular tone? Line 131 – address how neuronal NOS contributes to blood flow.

L250 –include which “vascular regions” are affected.  Add “such as …” 

Table 1 is unnecessary.  It does, however, address the changes in local tissue metabolism that could contribute to ontogenetic changes in blood flow but this should be included in the text.  Further, if this table is included, it should include the references that identify the role of eNOS in the specific organ beds.

Author Response

Reply to Reviewer 1

We would like to thank this Reviewer for the positive evaluation of our manuscript. Our point-by-point responses to the questions and concerns are below. The changes are highlighted by red color in the main text of the manuscript.

Comments and Suggestions for Authors

Summary:  This review evaluates the literature on the ontogenesis of endothelium-derived NO in regulating organ blood flow in different circulations.  Further, the role of thyroid hormone during pregnancy is discussed in the context of its regulation of NO during early neonatal and adult stages of maturation.  It concludes that since thyroid hormone plays an important role in NO production that the alteration in thyroid hormone levels (i.e. during hypo and hyperthyroidism) in pregnant women is an important regulator of organ blood flow and systemic blood pressure in the early vs late offspring.

Strengths – The review is very well written and covers specific topics in a clear and organized manner.  It does a good job of addressing the role of eNOS in organ-specific circulations and the ontogenetic changes that occur in blood vessels as well as the possible underlying mechanisms that contribute to the changes in eNOS activation with age.  The role of thyroid hormone as an important regulator of NO production and, hence, vascular tone in conditions of hypo vs hyperthyroidism is well made and convincing.  Further, the case is well made as to the effect of maternal hypo/hyperthyroidism on programming effects of vascular responses in the offspring.

Concerns to address –

1. The coronary circulation is an important vascular bed that should be included in a separate paragraph.  It could still be included with regards to the influence of thyroid levels as in the latter sections.  It might be important to address other endothelium-derived factors such as epoxyeicosatrienoic acids as endothelium-derived hyperpolarizing factors, particularly in the coronary circulation.  Lastly, there is a very nice review on the role of thyroid hormone in development of the fetal heart (SS Jonker and S Louey, J Endocrinol., 2016).  While it does not address endothelium-dependent mechanisms it may play into your thoughts regarding thyroid’s developmental role in the coronary circulation.

Our response:

According to your suggestion, the available data on developmental changes in NO contribution to vascular tone regulation in the coronary circulation were included into the main text of the manuscript (lines 106-118). Regarding possible developmental alterations of the EDHF-pathway, we did not address it in any other section of our review. This review is focused solely on eNOS-derived NO. We agree that other endothelium-derived factors are important, but this is a large topic beyond the scope of our manuscript and should be addressed in a separate manuscript. The review on the role of thyroid hormone in development of the fetal heart (SS Jonker and S Louey, J Endocrinol., 2016) was cited in the text.

While the focus of this article is on NO’s role in mediating vascular reactivity, when discussing vascular tone and blood flow and the ontogenetic changes that occur, the role of local tissue metabolism is not mentioned.  It should be included that in certain organs such as skeletal muscle and intestine changes in vascular tone are highly influenced by its metabolic rate and motility, respectively.  Thus, while changes in eNOS activity may contribute to baseline levels of tone, there may be other factors that contribute to ontogenetic changes in blood flow such as tissue metabolism rather than to NO release only. 

Our response:

Thank you very much for pointing to this issue. Undoubtedly, the regulation of vascular tone is performed by multiple factors and NO is only one of them. Along with that different factors can interact and influence each other's effects. For example, an intensification of tissue metabolism can increase eNOS activity by either interstitial acidification (smaller intramuscular arteries - (Celotto et al., 2011)) or an increase of shear stress (in larger feed arteries - (Balligand et al., 2009)). This information has been added to the text (lines 211-216).  

Another mechanism of reduced NO influence with age could be related to a maturational increase in the number of vascular smooth muscle layers from early to late adult stages.  Thus, NO may have a lesser influence on artery walls with thicker versus thinner thickness due to the number of layers and a lesser influence on vascular tone.  This was marginally addressed on Line 55 but could be addressed more directly.

Our response:

Thank you very much for your comment. We agree that a reduced NO influence with age might be related to a maturational increase in the number of vascular smooth muscle layers from early to late adult stages. But we do not believe it to be the main mechanism reducing the functional activity of NO-signaling pathway during maturation. This issue is discussed now in the review (lines 208-211).

Information on thyroid levels would be helpful.  This information, if available, would help put the role of thyroid hormones in a physiological vs pathological context. 

Our response:

The information on the changes of thyroid hormone levels during development was added to the text of the review (lines 236-239).

Renal circulation – Line 130 –clarify “contribution of NO to vascular tone” – do you mean in reducing vascular tone? Line 131 – address how neuronal NOS contributes to blood flow.

Our response:

Corrected (lines 155 and 156-157, respectively).

L250 –include which “vascular regions” are affected.  Add “such as …” 

Our response:

The specification of vascular regions was added to the text of the review (lines 288-289).

Table 1 is unnecessary.  It does, however, address the changes in local tissue metabolism that could contribute to ontogenetic changes in blood flow but this should be included in the text.  Further, if this table is included, it should include the references that identify the role of eNOS in the specific organ beds.

Our response:

In our opinion, Table 1 summarizes the developmental alterations in the NO-dependent regulation of arterial tone in different vascular regions. Thus, we would like to leave it in the review. According to your suggestion we included the references into this table.

Reviewer 2 Report

The review is wonderful and I have had a great time reading it.

 Although I think that the review is well constructed and prepared, I would like to see, probably a table that summarizes changes in NO in the different stages of development of humans and/or animals. That would be very beneficial. That could include NOS activity measured in several organs at different stages of ontogenesis.

Are there studies about the role of thyroid hormones stimulating eNOS or iNOS?

How about experimental models of thyroid deficiency and L-NAME supplementation in rats or mice during prenatal life?

Author Response

Reply to Reviewer 2

We would like to thank this Reviewer for the positive evaluation of our manuscript. Our point-by-point responses to the questions and concerns are below. The changes are highlighted by red color in the main text of the manuscript.

Comments and Suggestions for Authors

The review is wonderful and I have had a great time reading it.

Although I think that the review is well constructed and prepared, I would like to see, probably a table that summarizes changes in NO in the different stages of development of humans and/or animals. That would be very beneficial. That could include NOS activity measured in several organs at different stages of ontogenesis.

Our response:

We agree, that it would be nice to have such a table. However, this table cannot be prepared on the basis of the currently available literature. Thus, different groups reported data obtained from different species at different developmental stages. Considering the well-known variability in maturation rates in different mammalian species, it is very difficult to correlate the data with defined stages of development across species. At the moment it has to be concluded that considerably more data are needed to get a general, i.e. across species, picture of the changes of the contribution of NO to vascular tone regulation at different stages of development.

Are there studies about the role of thyroid hormones stimulating eNOS or iNOS?

Our response:

Data about the role of thyroid hormones stimulating eNOS expression/activity are discussed in Section 3.1.

Regarding iNOS, we have never detected its mRNA in resistance arteries from either healthy rats or rats with antenatal/early postnatal hypothyroidism (Gaynullina et al., 2013, 2017, Sofronova et al., 2016, 2017). Along with that, increases in iNOS expression levels were shown in aortic and heart samples from hyperthyroid rats (thyroxine administration for 6 weeks) (RodrĂ­guez-GĂłmez et al., 2016) as well as in heart samples from adult rats with fetal hypothyroidism (Jeddi et al., 2016). In this review, however, we focused on the maturational changes and hormonal regulation of eNOS and, therefore, did not discuss these topics in relation to iNOS. 

How about experimental models of thyroid deficiency and L-NAME supplementation in rats or mice during prenatal life?

Our response:

Prenatal thyroid deficiency was studied by our group using the model of antenatal/early postnatal hypothyroidism, when dams were treated with antithyroid drugs from the first day of pregnancy till 2 weeks after delivery. The vascular effects of this model in 2-week-old offspring are described in Section 3.2 and in the paper by Sofronova et al., 2017. According to your comment, we clarified the described model (antenatal/early postnatal hypothyroidism) in the text of our review (line 281).

According to the literature, L-NAME administration to pregnant dams causes hypertension in their adult offspring  (Tain et al., 2015), but possible alterations of eNOS expression/activity were not addressed in this study.